# Genetic Diversity and Phylogenetic Relationships of *Castor fiber birulai* in Xinjiang, China, Revealed by Mitochondrial *Cytb* and D-loop Sequence Analyses

**DOI:** 10.3390/ani15142096

**Published:** 2025-07-16

**Authors:** Linyin Zhu, Yingjie Ma, Chengbin He, Chuang Huang, Xiaobo Gao, Peng Ding, Linqiang Zhong

**Affiliations:** 1Xinjiang Key Laboratory of Biological Resources and Genetic Engineering, School of Life Science and Technology, Xinjiang University, Urumqi 830000, China; zhujulien00@gmail.com (L.Z.);; 2Xinjiang Burgen Beaver National Nature Reserve, Qinghe 836200, China; 3Ecological and Geographical Engineering Center, Xinjiang Geological Engineering Company, Urumqi 830000, China; 4Wildlife Conservation Association of Xinjiang Uygur Autonomous Region, Urumqi 830000, China; 5Xinjiang Key Laboratory for Ecological Adaptation and Evolution of Extreme Environment Biology, College of Life Sciences, Xinjiang Agricultural University, Urumqi 830000, China

**Keywords:** *Castor fiber birulai*, genetic diversity, lineage independence, mitochondrial cytochrome b, D-loop sequences

## Abstract

*Castor fiber birulai* is one of the rarest lineages of the Eurasian beaver and is found only in the Buergen River Basin of Xinjiang, China. Compared with beaver populations in Europe and Russia, its numbers are small, and it has been relatively understudied. To better understand the genetic status of this endangered population, we analyzed genetic samples from 19 individuals. Our results showed that the genetic diversity of the beavers in China is low, with close relationships between individuals. Compared to other Eurasian populations, they exhibited clear genetic differences, suggesting that this group may represent a distinct conservation unit. Although geographically close to beavers in Mongolia, gene flow between the two populations appears to have been disrupted by human activities such as dam construction and border fencing. Based on these findings, we recommend strengthening cross-border cooperation between China and Mongolia to protect the habitat of this rare species. We also suggest including nuclear DNA in future studies to fully assess its conservation value. This research provides important scientific support for ecological protection and biodiversity conservation in border regions.

## 1. Introduction

The Eurasian beaver (*Castor fiber*), one of only two extant species within the genus Castor, was historically widespread across the Eurasian continent. As a keystone species and ecological engineer in riparian ecosystems, it plays a pivotal role in maintaining wetland structure, regulating hydrological processes, and enhancing biodiversity [1]. However, since the Middle Ages, the population of *C. fiber* has declined dramatically due to intensive fur trade, overhunting, and habitat destruction. By the early 20th century, the global population had dwindled to fewer than 1200 individuals, surviving in only eight geographically isolated refugia across France, Norway, Germany, Belarus, and Russia [2].

With the implementation of wildlife protection policies and the advancement of reintroduction programs, wild *C. fiber* populations have been reestablished in more than 25 countries. The global population has now recovered to over 1.5 million individuals [3,4,5,6]. Despite this remarkable recovery, concerns have emerged regarding lineage admixture, genetic homogenization, and reduced local adaptation associated with artificial translocations. These genetic and ecological challenges pose significant risks to the long-term viability and sustainability of reintroduced populations [7,8,9].

Studies have shown that the extant maternal lineages of *C. fiber* can be broadly classified into two major clades: a western lineage (populations in France, Germany, and surrounding regions) and an eastern lineage (populations from Russia, Mongolia, and adjacent areas), reflecting their independent evolutionary trajectories shaped by isolation in glacial refugia [10]. On a global scale, naturally distributed populations of *C. fiber* that retain their original lineage without anthropogenic disturbance are exceedingly rare. *Castor fiber birulai*, native to the Ulungur River Basin in Xinjiang, China, represents the only indigenous subspecies of *C. fiber* in the country and is one of the few remaining subspecies worldwide with a genetically intact and evolutionarily continuous lineage. As one of the descendants of the eight historical refugia of *C. fiber* [3], *C. f. birulai* occupies a clearly defined geographical range and has been geographically isolated for a long period of time. Its core habitat lies in the Ulungur River Basin, including its upper tributaries in China and the upstream sections of the Bulgan and Tes Rivers in Mongolia [11,12]. The Bulgan River, a cross-border watercourse between China and Mongolia, has historically facilitated the natural distribution of the species across both countries.

Recent population estimates suggest that approximately 800 individuals remain in China and approximately 500 individuals remain in Mongolia, with the total number of *C. f. birulai* individuals falling below 1300 individuals. The subspecies is characterized by a narrow geographic distribution, a small and endangered population size, and pronounced habitat fragmentation and isolation.

*C. f. birulai* is recognized as a core species of the eastern lineage of Eurasian beavers and is one of the eight primary phylogeographic lineages proposed by Ducroz et al. [10]. Owing to its long-term isolation in closed river systems in arid environments, this subspecies has remained largely free from interlineage admixture and is therefore considered a classic example of a “genetic relict population” with high lineage integrity and evolutionary continuity [13]. On the basis of complete mitochondrial genome analysis, Horn inferred that the major divergence events among Eurasian beaver matrilines likely occurred approximately 210,000 years ago [14]. It is plausible that *C. f. birulai* has retained ancestral characteristics from these early divergences, underscoring its evolutionary significance and conservation value.

Nevertheless, systematic research on the genetic diversity, population structure, and phylogenetic placement of *C. f. birulai* remains remarkably insufficient. In contrast to the comprehensive multilocus genetic assessments conducted for many reintroduced populations in Europe, research on *C. f. birulai* has long been marginalized both domestically and internationally. The existing data are sparse and lack adequate geographical representation, thereby limiting the scientific basis for effective conservation management of this subspecies.

To address this gap, representative samples of *C. f. birulai* from the Buergen River Basin in Xinjiang, China, were collected to conduct a systematic evaluation of haplotype diversity, phylogenetic structure, and demographic history using mitochondrial cytochrome b (*cytb*) and control region (D-loop) sequences in this study. By integrating these data with publicly available sequences from *C. fiber* subspecies across Eurasia, we aimed to clarify the phylogenetic placement and evolutionary distinctiveness of *C. f. birulai* and highlight the importance of its conservation. These findings are intended to provide a scientific foundation and technical support for the development of targeted genetic resource management and conservation strategies for this taxon.

## 2. Materials and Methods

### 2.1. Sample Collection

Between 2023 and 2025, a total of 19 genetic samples of *C. f. birulai* were collected from the Buergen River Basin. All the samples were collected from different individuals to ensure proper representation of the population genetic structure. Of these, 12 faecal samples were obtained from rescued individuals housed within the Xinjiang Buergen River Beaver National Nature Reserve, whereas 7 muscle tissue samples were obtained from naturally deceased individuals preserved in the Tianma Natural History Museum, Xinjiang (see Appendix A). These sampling sites encompass the major distribution area of *C. f. birulai* within China, offering strong geographic representation and providing a solid foundation for subsequent analyses of genetic structure and lineage differentiation.

Following collection, all the tissue samples were immediately stored in liquid nitrogen and subsequently transferred to a laboratory facility, where they were preserved at −80 °C until DNA extraction. The sampling procedures adhered to ethical standards and technical guidelines for genetic research on wildlife and were approved under the relevant ethics review protocol (XJUAE-2025-018).

### 2.2. DNA Extraction and Amplification of Cytb and D-Loop Genes

Genomic DNA was extracted from beaver faecal samples using the QIAamp Fast DNA Stool Mini Kit (Qiagen, Hilden, Germany) following the manufacturer’s protocol. Muscle tissue DNA was extracted using the Ezup Column Animal Genomic DNA Purification Kit (Sangon Biotech, Shanghai, China) according to the manufacturer’s instructions.

On the basis of the published complete mitochondrial genome of the Eurasian beaver, primers for *Cytb* gene amplification were designed using Primer Premier v5.0 [15]. The forward primer was 5′-TCGACCTCCCAACACCAT-3′, and the reverse primer was 5′-GAGAGCTACTACTCCACCGA-3′. Amplification of the mitochondrial D-loop region was performed using the primers Thr-L15926 (5′-CAATTCCCCGGTCTTGTAAACC-3′) and DL-H16340 (5′-CCTGAAGTAGGAACCAGATG-3′), following protocols described by Ducroz and Durka [3,10].

A reaction mixture (25 μL) containing 12.5 μL of 2× PCR buffer, 5 μL of dNTPs (2 mmol/L), 1 μL of KOD FX NEO polymerase (1.0 U/μL), 0.75 μL of each primer (10 pmol/μL), 1.5 μL of template DNA, and 3.5 μL of nuclease-free water was prepared for PCR. The thermal cycling conditions were as follows: initial denaturation at 96 °C for 4 min; 35 cycles of denaturation at 96 °C for 40 s, annealing at 56 °C for 45 s, and extension at 72 °C for 1 min; and a final extension at 72 °C for 10 min and storage at 4 °C.

The amplification products were assessed via 1.0% agarose gel electrophoresis to verify the fragment size and concentration and were stored at −20 °C prior to sequencing. The PCR products of the mitochondrial *Cytb* and D-loop regions were sequenced by Sangon Biotech (Shanghai, China). The resulting DNA sequences were verified and annotated through BLAST (version 2.14.1.) searches against the GenBank database.

### 2.3. DNA Sequence Data Processing

Bidirectional sequencing chromatograms of the *Cytb* and D-loop genes were initially subjected to manual inspection and correction using Chromas (version 2.6.6) and EditSeq (version 15.3.0) software to ensure sequence accuracy and consistency [16]. Subsequently, sequence trimming and assembly were conducted employing the SeqMan module of the DNAStar 11.0 software suite [17]. The resulting sequences were aligned with the ClustalW algorithm implemented in MEGA 11.0 to produce a standardized alignment. Base composition and the ratio of transitions (Ti) to transversions (Tv) were calculated using MEGA 11.0 for both *Cytb* and D-loop sequences to assess mutational bias and inform the selection of substitution models. Phylogenetic analyses were conducted in MrBayes v3.2.7, where separate Bayesian inference trees were reconstructed for the *Cytb* and D-loop datasets, and a concatenated analysis was also performed. The GTR + I + G substitution model was employed, allowing independent estimation of substitution rates among nucleotide pairs while accounting for base frequency differences, transition/transversion bias, the proportion of invariable sites, and among-site rate variation. Four Markov chains were run for 1,000,000 generations, sampling every 100 generations, with the first 25% discarded as burn-in. Posterior probabilities were used to assess clade support [18].

The haplotype number (h), haplotype diversity (Hd), and nucleotide diversity (π) were estimated for the entire dataset and for each geographic population using DnaSP 6 [19]. Mismatch distribution analyses were performed to evaluate historical demographic expansions. Neutrality tests, including Tajima’s D [20] and Fu’s Fs [21], alongside pairwise genetic differentiation tests (Fst) [22], were performed using Arlequin 3.1 [23]. Statistical significance was assessed through 1000 bootstrap replicates [24].

To contextualize the genetic data of *C. f. birulai*, a comprehensive mitochondrial dataset was compiled from the GenBank database, encompassing global *C. fiber* and *Castor canadensis* sequences. Specifically, 16 *Cytb* and 92 D-loop sequences of *C. fiber*, along with 4 *Cytb* and 14 D-loop sequences of *C. canadensis*, were obtained for phylogenetic reconstruction, haplotype network analysis, and genetic distance estimation. For phylogenetic analyses, the European otter (*Lutra lutra*) was designated the outgroup (see Appendix A).

For the analysis of population demographic history, four major geographic units were defined: Russia (two *C. fiber* subspecies from Russia), China (*C. f. birulai* beaver populations from China and Mongolia), Europe (three *C. fiber* subspecies from Europe), and North America (representative populations of the North American beaver (*Castor canadensis*) (see Appendix A).

## 3. Results

### 3.1. Mitochondrial Sequence Analysis

High-quality genomic DNA was successfully extracted from all 19 *Castor fiber birulai* samples, and clear amplification bands were obtained for both the *Cytb* and D-loop regions, indicating successful PCR amplification (see Appendix A). After alignment and manual correction, the *Cytb* sequence was 800 bp in length, with the following base composition: adenine (A) 24.5%, thymine (T) 31.1%, cytosine (C) 31.3%, and guanine (G) 13.1%. The G content was the lowest among the four bases, and the combined A + T content (55.6%) exceeded that of C + G (44.4%), indicating a marked AT bias typical of vertebrate mitochondrial DNA.

The aligned D-loop sequences were 592 bp in length, with the following base compositions: A 27.1%, T 32.3%, C 15.7%, and G 24.8%. The A + T content was 59.4%, again exceeding the G + C content (40.5%) and reflecting a similar AT bias. Substitutions in this region were predominantly transitions, with an estimated transition/transversion ratio (R) of 1.77, indicating a notable mutational bias. This is consistent with typical mitochondrial DNA substitution patterns and supports the use of substitution models (e.g., GTR) that account for Ti/Tv bias in phylogenetic reconstruction.

### 3.2. Genetic Diversity of C. f. birulai

The analysis of the 19 samples revealed a single haplotype in the *Cytb* gene region. In contrast, three haplotypes were identified in the D-loop region, with Hap_1 being the predominant haplotype. The haplotype diversity (Hd) of the D-loop region was 0.444, the nucleotide diversity (π) was 0.0043, and the mean number of nucleotide differences (K) was 2.56.

The *Cytb* and D-loop sequences were concatenated to generate a combined sequence of 1258 bp in length. Three haplotypes were identified from the concatenated dataset, which was consistent with those detected in the D-loop region alone. The diversity indices for the concatenated sequence were as follows: Hd = 0.444, π = 0.00181, and K = 2.281 (see Table 1).

### 3.3. Genetic Structure of Xinjiang C. f. birulai

The D-loop region of the *C. f. birulai* population in China presented low genetic diversity, with a haplotype diversity (Hd) of 0.576 and nucleotide diversity (π) of 0.00515. In contrast, only a single haplotype was detected in the *Cytb* region, and no nucleotide diversity was detected, indicating an extremely low level of genetic variation within this mitochondrial locus.

In comparison, the Mongolian *C. f. birulai* population presented slightly greater genetic diversity (D-loop: Hd = 0.600, π = 0.00248; *Cytb*: Hd = 0.667, π = 0.00083). However, the overall genetic diversity of both Chinese and Mongolian populations remains significantly lower than that reported in other geographic populations of *C. fiber* and *C. canadensis* (see Table 2).

### 3.4. Haplotype Network Analysis

Haplotype network analysis revealed a total of eight *Cytb* haplotypes in beavers distributed across six countries and 52 D-loop haplotypes in beavers distributed across 12 countries (Figure 1). The *C. fiber* populations presented pronounced geographic genetic structuring. Haplotype networks based on both *Cytb* and D-loop sequences consistently indicated substantial genetic differentiation between *C. f. birulai* and other Eurasian beaver subspecies distributed in Europe.

In contrast, European beaver populations presented greater haplotype diversity and a more complex phylogeographic structure. Notably, the Russian beaver samples formed several localized clusters, suggesting diverse historical migration routes. The Romanian beaver samples occupied a central position within the network, linked multiple geographic groups, and may have functioned as a genetic bridge during the historical dispersal of *C. fiber* across Eurasia.

Phylogenetic trees constructed on the basis of *Cytb*, D-loop, and concatenated mitochondrial sequences consistently demonstrated that samples from the China–Mongolia border region belong to the subspecies *C. f. birulai*, forming a stable, distinct clade characterized by strong geographic clustering and genetic homogeneity. In contrast, samples from European populations formed a separate and more complex clade, indicative of a polyphyletic origin (Figure 2). Within this European clade, individuals from Germany, Norway, Austria, and Poland presented greater genetic variation, reflecting multiple historical reintroduction events and admixture from diverse sources. Notably, the British samples grouped with those from Central Europe to form a relatively distinct monophyletic lineage.

Analyses of genetic distance and the genetic differentiation index (Fst) among Eurasian beaver populations revealed minimal genetic distance (0.00269) between the Chinese and Mongolian *C. f. birulai* populations (Figure 3). However, the corresponding Fst value was notably high (0.67055), suggesting the presence of significant population structure and limited gene flow between the two national populations. Furthermore, pairwise Fst values between Chinese *C. f. birulai* and various European subspecies—including *C. f. albicus* (Germany), *C. f. galliae* (France), *C. f. fiber* (Norway), and *C. f. pohlei*, *C. f. tuvinicus*, and *C. f. belorussicus* (Russia)—exceed 0.95, indicating a high degree of genetic differentiation between *C. f. birulai* and other European lineages (Table 3).

### 3.5. Demographic History Analysis

Neutrality tests and mismatch distribution analyses based on the D-loop region revealed distinct demographic patterns among geographic groups (Figure 4). For both the Chinese and European populations, the values of Tajima’s D and Fu’s FS were positive but not statistically significant (China = 0.19152; EU = 1.11565), suggesting an absence of recent population expansion or contraction. The mismatch distribution curve for the Chinese population exhibited a relatively smooth and unimodal shape, closely matching the expected distribution under demographic equilibrium, which is consistent with a stable population history.

In contrast, the North American population presented a significantly negative Fu’s FS value (−5.95), whereas the Russian population also presented a negative Fu’s FS value (−2.71), both of which are indicative of historical demographic expansion. The multimodal shape observed in the mismatch distribution of the Russian population further supports the hypothesis of past expansion events, potentially associated with internal genetic substructure or secondary contact following historical range shifts (see Appendix A).

## 4. Discussion

Genetic diversity is closely linked to a species’ ability to adapt to new environments and survive over evolutionary time. It is a key indicator in assessing the long-term viability of populations, as its loss can lead to decreased individual fitness and reproductive success. In this study, the Chinese population of *C. f. birulai* presented extremely low genetic variation in both the mitochondrial *Cytb* and D-loop markers. The number of haplotypes and diversity indices were significantly lower than those reported in other populations of *C. fiber* and *C. canadensis*, indicating an overall reduction in genetic diversity. As a protein-coding region of the mitochondrial genome, *Cytb* is relatively conserved and has low mutation rates, often presenting limited polymorphisms or even a single haplotype in small populations [25,26]. In contrast, the D-loop, as a noncoding control region, evolves faster and accumulates mutations more readily, thus providing richer haplotype information within populations [27].

Previous studies have shown that small and isolated populations of endangered wildlife often present low genetic diversity, habitat-specific adaptations, and loss of migratory behaviour [28]. For example, the isolated population of the Galápagos giant tortoise (*Chelonoidis abingdonii*) presented extremely low genetic diversity, with certain genetic regions lacking haplotype variation. This genetic homogenization severely constrains the adaptive potential of the population [29]. Similarly, the low census size, narrow distribution, and fragmented habitats of *C. f. birulai* may restrict gene flow and impede the accumulation and maintenance of novel genetic variation.

Haplotype network and phylogenetic analyses revealed that *C. f. birulai* constitutes a highly distinct lineage that is clearly differentiated from other European *C. fiber* populations. This divergence reflects not only the evolutionary trajectory of a peripheral population under strong genetic drift but also the role of geographic isolation and ecological compartmentalization as barriers to gene flow [10]. Similar phylogenetic isolation has been reported in other endangered species with restricted ranges, such as the Hainan gibbon (*Nomascus hainanus*) and Yunnan snub-nosed monkey (*Rhinopithecus bieti*), both of which exhibit lineage conservatism and low genetic diversity due to geographic and ecological isolation [30,31]. In the populations of Eurasian otters (*Lutra lutra*) living in the Pyrenean edge zone, mountainous terrain and isolated river systems likewise hinder gene flow and result in a loss of genetic diversity [32]. These patterns across species underscore the critical role of ecological and geographic barriers in driving lineage independence and reducing genetic diversity in endangered populations.

In contrast, European *C. fiber* populations present high lineage admixture due to extensive anthropogenic reintroduction during the 20th century. Owing to a lack of lineage-level assessment during release programs, individuals of various subspecies and geographic origins were often translocated into the same areas. For example, individuals from western Europe (*C. f. albicus*), France (*C. f. galliae*), Norway (*C. f. fiber*), and the Voronezh breeding center in Russia (*C. f. orientoeuropaeus*) were all introduced into regions of Germany, Poland, Austria, and Switzerland. This has led to the emergence of hybrid zones where lineage integrity is significantly disrupted. Although such genetic admixture may increase short-term diversity and promote population growth, it can also undermine local adaptations, lead to behavioural changes, and alter micro-habitat utilization strategies, raising concerns about “adaptive conflict” and “genetic pollution” [33,34]. In Austria and Switzerland, maternal lineage disruption and lineage replacement have already been observed in reintroduced populations [35,36]. Prolonged admixture may further erode rare lineage-specific traits, compromising species-wide phylogenetic integrity and evolutionary resilience [37]. A similar controversy arose in the Alpine brown bear reintroduction project, where mixed-lineage individuals led to the extinction of local lineages [38]. Unlike the multiple lineage admixture observed in European Eurasian beaver populations, the Mongolian–Xinjiang beaver represents a highly distinct and genetically unintrogressed relic lineage. Therefore, its conservation and management should prioritize the preservation of its native phylogenetic integrity.

Therefore, future reintroduction efforts must prioritize subspecies and lineage integrity. Molecular evaluations should precede releases to ensure compatibility with existing lineages in target habitats. For example, the “lineage-priority model” implemented in European brown bear (*Ursus arctos*) management requires strict geographic matching to prevent genetic dilution [39]. In China, reintroduction programs for the forest musk deer (*Moschus berezovskii*) and Przewalski’s horse (*Equus przewalskii*) have likewise emphasized lineage recognition and isolation, with enforced “lineage consistency” standards, particularly for cross-border population reinforcement [40,41]. Taken together, *Castor fiber birulai* exhibits high mitochondrial lineage distinctiveness, clear geographic isolation, and significant genetic differentiation, all of which underscore its considerable conservation value. Management efforts should therefore prioritize the preservation of its unique phylogenetic characteristics, and the introduction of non-native lineages during reintroduction or ex situ conservation programs should be approached with caution to maintain its native genetic structure. According to the framework of Evolutionarily Significant Units (ESUs) proposed by Moritz [25], our current assessment—based solely on mitochondrial genetic data—is not sufficient to fully evaluate the evolutionary independence of this population. In future work, we plan to expand the sampling range and increase sample size, integrating allele frequency data from nuclear loci to comprehensively assess the historical genetic structure of the population and determine whether *C. f. birulai* meets the criteria for ESU designation. Geue and Crandall [42,43] emphasized that the maintenance of ESUs contributes to ecosystem stability and resilience. As an ecological engineer within riverine ecosystems, *C. f. birulai* plays a critical role in hydrological regulation, wetland structure maintenance, and species coexistence. Therefore, conservation of *C. f. birulai* should not focus solely on its genetic distinctiveness, but also recognize its function as a keystone component in sustaining the ecological integrity of transboundary river systems. Its potential status as an ESU thus holds broad ecological and evolutionary significance.

Combined analyses of *Cytb* and D-loop sequences revealed that *C. f. birulai* individuals from China and Mongolia form a stable, distinct clade, exhibiting pronounced phylogenetic differentiation and maternal monophyly. Despite their small average genetic distance (0.00269), a high Fst value (0.67055) reflects significant genetic differentiation. These findings highlight the influence of physical barriers, habitat fragmentation, genetic drift, and founder effects in shaping divergence [44,45]. The Buergen River, a cross-border watercourse connecting China and Mongolia, has been affected by dam construction and habitat fragmentation, which likely impedes gene flow across the border. Antoine [45] emphasized that the loss of habitat connectivity is a powerful driver of rapid genetic differentiation, even in the absence of large geographic distances. For example, wild giant pandas (*Ailuropoda melanoleuca*) have formed distinct genetic clusters across Sichuan, Shaanxi, and Gansu Mountains because of fragmented forests and human barriers [46]. Similarly, the brown-eared pheasant (*Crossoptilon mantchuricum*) has shown significant genetic differentiation among populations in the Qinling–Taihang region because of deforestation and agricultural expansion [47]. Together, these cross-species findings illustrate a causal chain linking habitat fragmentation, loss of gene flow, and lineage divergence.

Historical demographic analyses further revealed that the *C. f. birulai* population has not experienced recent expansion or contraction, remaining relatively stable. In contrast, the Russian population showed signs of historical expansion. Population expansion typically requires continuous, suitable aquatic habitats, abundant resources, low anthropogenic disturbance, and strong dispersal capacity. However, *C. f. birulai* inhabits arid river basins with limited water resources and geographic barriers, which constrain migration and expansion [48]. Although European populations have expanded in number and distribution, mitochondrial data have not mirrored this trend, possibly because of sample admixture, maternal lineage conservatism, and the delayed changes in the D-loop sequence in response to demographic changes [27]. Similar discrepancies have been reported in reintroduced forest musk deer [40], highlighting the need to incorporate nuclear genomic data and multi-generational monitoring to assess the genetic impact of population expansion accurately.

In contemporary conservation practice, endangered species management often faces a dilemma between “population recovery” and “genetic purity.” As one of the most phylogenetically distinct and genetically intact *C. f. fiber* subspecies, *C. f. birulai* serves as a living relic for understanding Castor evolution. This study also provides a valuable model for exploring the trade-off between phylogenetic conservatism and genetic diversity in small-range populations.

## 5. Conclusions

This study provides the first comprehensive assessment of the mitochondrial genetic diversity and phylogenetic relationships of *Castor fiber birulai* in Xinjiang, China. Analyses of *Cytb* and D-loop sequences revealed that this population exhibits extremely low genetic diversity and forms a highly distinct maternal lineage that is clearly differentiated from other *Castor fiber* subspecies across Eurasia. Although the populations in China and Mongolia are geographically proximate, they show significant genetic differentiation, likely driven by historical isolation, habitat fragmentation, and anthropogenic disruption of gene flow. These findings suggest that *C. f. birulai* is a relict population with high evolutionary distinctiveness and may possess characteristics consistent with an Evolutionarily Significant Unit (ESU), although further research is needed to confirm this designation.

Given its genetic uniqueness, narrow geographic distribution, and critical role as an “ecosystem engineer” in riverine systems, *C. f. birulai* should be treated as a high-priority conservation target. Management efforts should emphasize the preservation of its native phylogenetic integrity and exercise caution against the introduction of non-native lineages during reintroduction or ex situ conservation. Although this study confirms its genetic distinctiveness based on mitochondrial data, further studies incorporating broader geographic sampling and nuclear genomic data are needed to evaluate its population history and evolutionary independence, thereby providing a more robust basis for ESU recognition. Conserving *C. f. birulai* is not only crucial for preserving a rare genetic lineage but also essential for maintaining the ecological stability and resilience of transboundary river systems in Central Asia, underscoring the broader ecological and evolutionary significance of its potential ESU status.

## Figures and Tables

**Figure 1 animals-15-02096-f001:**
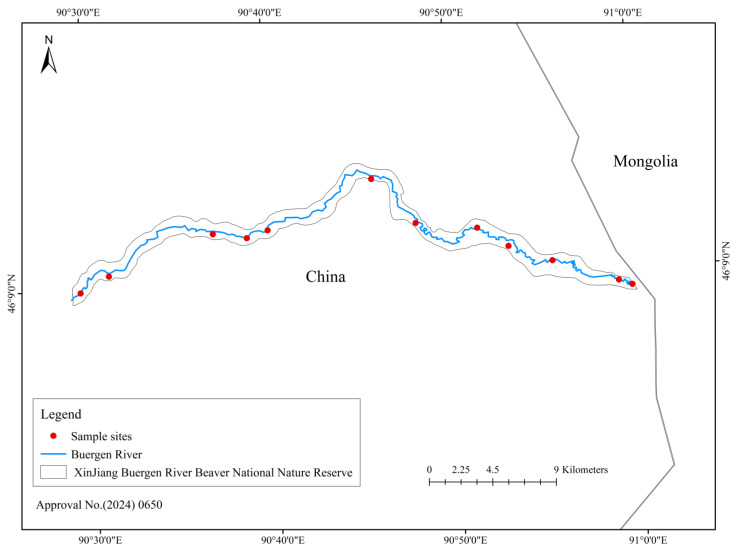
Distribution of *Castor fiber birulai* sampling sites along the Buergen River Basin in Xinjiang, China.

**Figure 2 animals-15-02096-f002:**
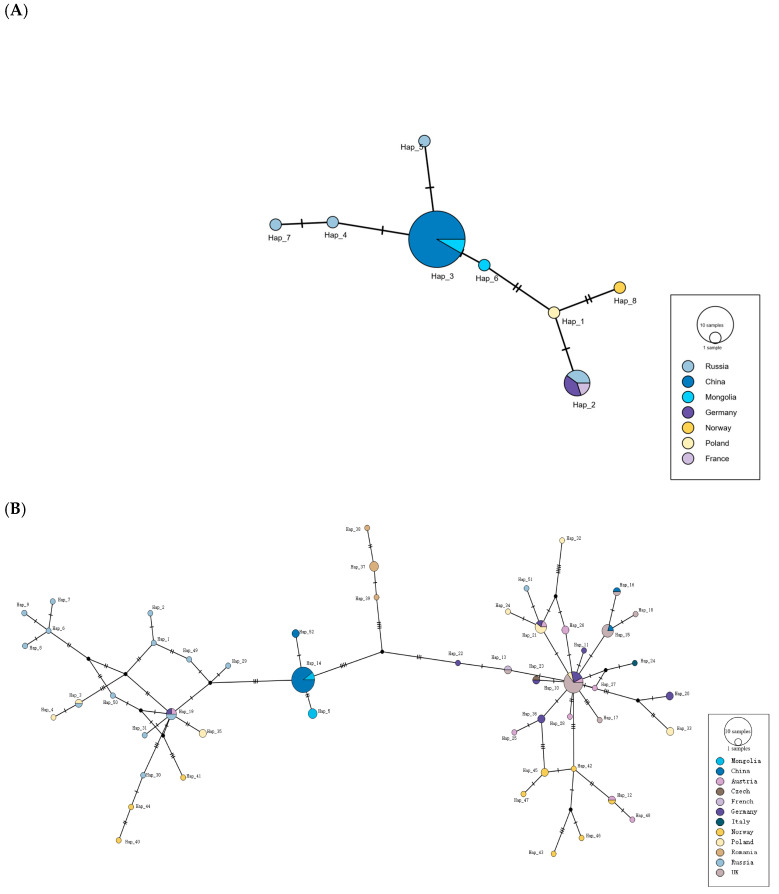
Haplotype networks of Eurasian beavers based on mitochondrial *Cytb* (**A**) and D-loop (**B**) sequences.

**Figure 3 animals-15-02096-f003:**
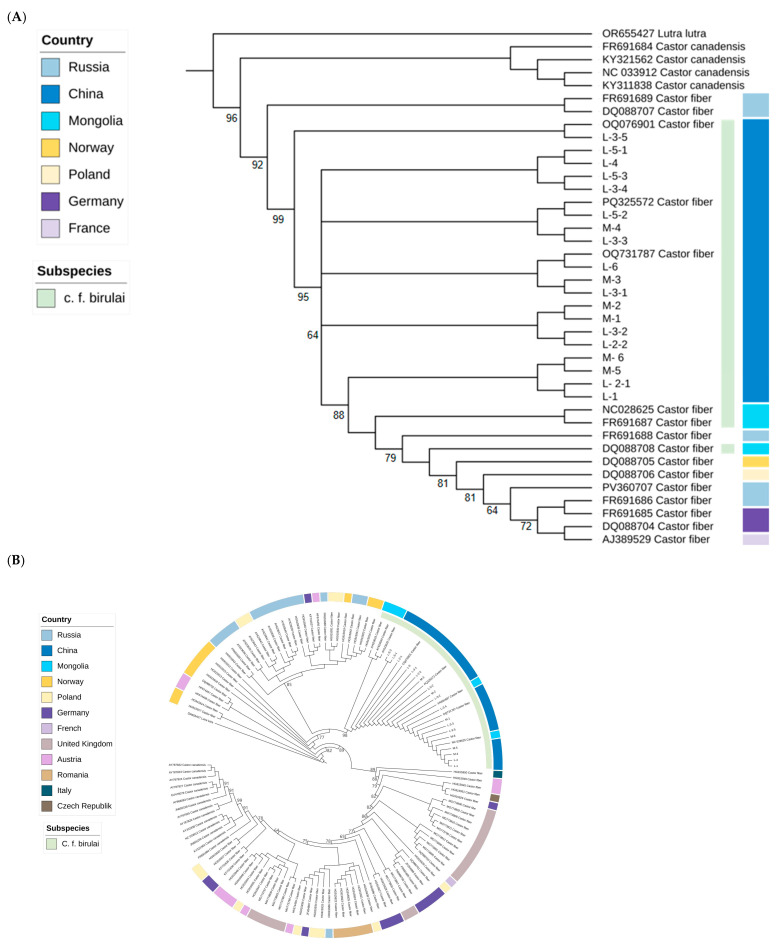
Phylogenetic trees of beavers based on *Cytb* (**A**), D-loop (**B**), and concatenated sequences (**C**). Node labels indicate posterior probabilities derived from MrBayes analysis. Although the values are displayed as “bootstrap” in the figure due to iTOL software (version 6.9.2) labeling conventions, they represent Bayesian posterior probabilities, not bootstrap support values.

**Figure 4 animals-15-02096-f004:**
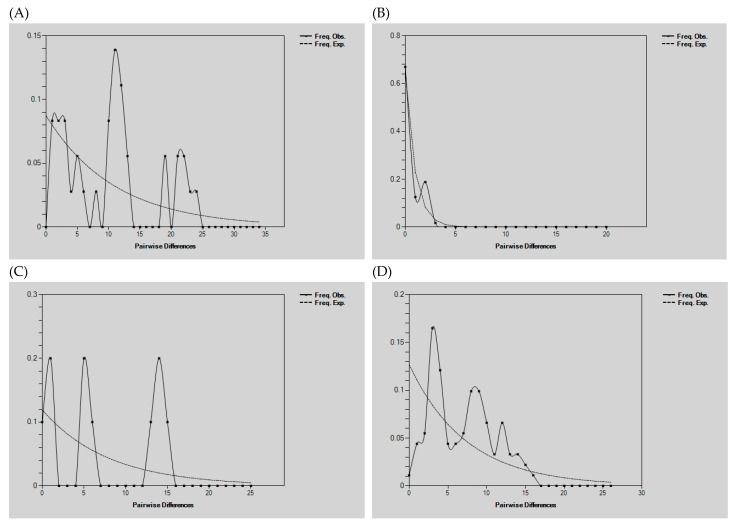
Mismatch distribution of D-loop haplotypes in Eurasian beavers: (**A**) Russia, (**B**) China, (**C**) Europe, and (**D**) *C. canadensis*.

**Table 1 animals-15-02096-t001:** Haplotype distribution of *C. f. birulai* (*Cytb*, D-loop, concatenated).

Genes	Bp	Haplotypes	Sample
*Cytb*	800	\	\
D-loop	592	Hap_1	L_1, L_2_1, L_2_2, L_3_1, L_3_2, L_3_3, L_3_4, L_3_5, L_4, L_5_2, M_2, M_3, M_4, M_5, M_6
		Hap_2	L_5_1, L_5_3
		Hap_3	L_6, M_1, M_2
*Cytb* + D-loop	1258	Hap_1	L_1, L_2_1, L_2_2, L_3_1, L_3_2, L_3_3, L_3_4, L_3_5, L_4, L_5_2, M_2, M_3, M_4, M_5, M_6
		Hap_2	L_5_1, L_5_3
		Hap_3	L_6, M_1, M_2

**Table 2 animals-15-02096-t002:** Genetic diversity of beavers by region (*Cytb*, D-loop, concatenated sequences).

Population	H		π		K	
	*Cytb*	D-loop	*Cytb*	D-loop	*Cytb*	D-loop
China-*C. f. birulai*	1	0.576	\	0.00515	\	2.41558
Mongolia-*C. f. birulai*	2	0.600	0.00083	0.00248	0.667	1.20
*Castor fiber*	7	0.967	0.00411	0.02123	3.289	8.15398
*Castor canadensis*	3	0.989	0.00440	0.01472	3.500	6.89011

**Table 3 animals-15-02096-t003:** D-loop genetic distance and Fst among beaver subspecies and regions.

Country	Subspecies	*C. f. pohlei*	*mogolin_* *C. f. briulai*	*C. f. tuvinivus*	*C. f. albicus*	*C. f. fiber*	*C. f. galliae*	*China_* *C. f. briulai*	*C. f. belorussicus*	*Castor canadensis*
Russia	*C. f. pohlei*	-	0.86900	0.95059	0.94203	0.89286	0.89831	0.97056	0.90323	0.93333
Mogolin	*mogolin_* *C. f. briulai*	0.02389	-	0.83434	0.95033	0.94059	0.93750	0.67055	0.93407	0.93627
Russia	*C. f. tuvinivus*	0.02357	0.02826	-	0.90330	0.84158	0.85965	0.95059	0.85321	0.93368
Germany	*C. f. albicus*	0.05007	0.04446	0.05262	-	0.95349	0.87500	0.98880	0.84615	0.93267
Norway	*C. f. fiber*	0.04022	0.04370	0.04367	0.03063	-	NA	0.99093	NA	0.92314
French	*C. f. galliae*	0.04248	0.04147	0.04958	0.01118	0.02770	-	0.99043	NA	0.92350
China	*China_* *C. f. briulai*	0.02494	0.00269	0.03002	0.04200	0.04125	0.03902	-	0.98987	0.96743
Russia	*C. f. belorussicus*	0.04474	0.03919	0.04727	0.00906	0.02990	0.01047	0.03676	-	0.91914
Canada	*Castor canadensis*	0.24048	0.21939	0.23211	0.22851	0.23189	0.23341	0.21630	0.21754	-

Note: Below the diagonal are genetic distances; above the diagonal are pairwise Fst values between populations.

## Data Availability

The mitochondrial DNA sequence data (*Cytb* and D-loop) generated during this study have been submitted to GenBank and are available under the accession numbers PV776086–PV776104 (*Cytb*) and PV776067–PV776085 (D-loop). These sequences are currently under embargo and will be publicly released upon publication of the article. Prior to release, the data are available from the corresponding author upon reasonable request.

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
