# Peer review of "Genetic Diversity and Phylogenetic Relationships of *Castor fiber birulai* in Xinjiang, China, Revealed by Mitochondrial *Cytb* and D-loop Sequence Analyses"

_animals, 2025, doi:10.3390/ani15142096_

Round 1

Reviewer 1 Report

Comments and Suggestions for Authors

The paper is clearly written, follows sound scientific methods, and adds new knowledge about the genetics of Castor fiber birulai, a beaver subspecies that has been little studied so far. The conclusions are well supported by the data, and the authors correctly highlight the importance of their findings for conservation—especially for managing Evolutionarily Significant Units (ESUs) and coordinating efforts between countries. I suggest accepting the manuscript after addressing minor revision.

This study fills a gap in genetic research on C. f. birulai, which is both genetically unique and geographically isolated. It uses two commonly applied mitochondrial DNA markers (Cytb and D-loop) to measure genetic diversity and differences between populations. By also including data from other regions, the study places C. f. birulai in a broader Eurasian context.

Please refer to the original article which stated “In contrast, European beaver populations presented greater haplotype diversity and a more complex phylogeographic structure. Notably, the Russian beaver samples formed several localized clusters, suggesting diverse historical migration routes. The Romanian beaver samples occupied a central position within the network, linked multiple geographic groups, and may have functioned as a genetic bridge during the historical dispersal of C. fiber across Eurasia”.

However, the study has some limitations:

It only uses mitochondrial DNA, which reflects maternal inheritance and cannot detect male-driven gene flow.

The sample size is relatively small (19 individuals), though this is understandable given the endangered status of the population.

Including nuclear DNA markers (like microsatellites or SNPs) would improve the understanding of population structure and history.

Future research should:

Include nuclear markers to complement the current findings and capture more genetic detail.

Consider landscape features (e.g., rivers, barriers) to better understand what limits movement and gene flow.

Collect more samples from both China and Mongolia to confirm the current patterns.

Discuss more clearly how the results could be used in practical conservation—such as habitat restoration, building wildlife corridors, or coordinating actions across borders.

Importantly, the authors should highlight how protecting C. f. birulai as an ESU can help preserve broader evolutionary processes and support healthy ecosystems. As noted by Geue et al. (2025, EcoEvoRxiv preprint), maintaining ESUs does not only protect one population—it also preserves genetic and ecological functions that benefit entire ecosystems.

A map showing the study area and the distribution of the analyzed samples would have been relevant in the introduction.

In table 3 (line 234) it is necessary to revise the names of the subspecies presented, so that they are correct (e.g. C. f. birulai instead of briulai; C.f. tuvinicus instead of tuvinivus).

In figure 3 we suggest that the title include the term "subspecies".

The species name can be highlighted by italics.

Author Response

We sincerely thank the reviewer for their valuable comments and suggestions. Please find below our detailed point-by-point responses.

Reviewer 2 Report

Comments and Suggestions for Authors

This manuscript provides an important contribution to the understanding of the evolutionary and genetic distinctiveness of Castor fiber birulai, a subspecies of high conservation interest. The authors use of mitochondrial markers to characterize lineage structure and divergence. Such an approach is appropriate, and the study fills a significant gap in current knowledge. However, while scientifically sound, the approach could use more refining, and would benefit from the use of nuclear markers, too, as mentioned here later.

The focus on C. f. birulai, a lineage largely absent from international research and conservation efforts, is original. The study highlights the need for lineage-level management and protection of peripheral, genetically distinct taxa. The study analyzes 19 individuals, which is reasonable given the small and endangered status of the population. However, the restricted geographic and numerical scope limits inference, particularly regarding demographic history and within-lineage diversity. The limited sample size and lack of spatial stratification constrain the ability to detect substructure and local differentiation. It is also important to have the inclusion of a map of sampling sites to provide spatial context.

The combination of mitochondrial Cytb and D-loop sequences allows for a clear identification of lineage-level divergence. The extremely low diversity found in Cytb (1 haplotype) and moderate variation in the D-loop region (3 haplotypes) is consistent with the expectations for an isolated and reduced population.

However, mitochondrial DNA reflects only the maternal lineage and lacks resolution on male-mediated gene flow or autosomal variation. Nuclear markers should be included to provide strength to the ms. This also applies to the invoked need to consider the pops as ESU based on genetic data. In fact, the most complete definition of ESU, based on genetic data is proposed by Moritz (1994), which emphasizes two criteria:i) reciprocal monophyly at mitochondrial loci and ii) significant divergence at nuclear loci.

I agree that in this manuscript, the authors demonstrate reciprocal monophyly at mitochondrial markers and high FST values with other C. fiber lineages, thus satisfying the first criterion. However, nuclear genetic differentiation is not assessed in this study, leaving the second ESU criterion unmet. This is an important limitation of this approach. Infact,  mitochondrial monophyly can arise from demographic processes such as bottlenecks or drift, rather than deep evolutionary independence.

The stronger recommendation for conservation, that is the designation of C. f. birulai as an ESU, is biologically plausible and well-supported by mitochondrial data, but incomplete under a strict Moritz framework due to the absence of nuclear evidence.

The authors may wish to include a critical reflection on these limitations in the discussion—acknowledging that ESU definitions have evolved (e.g., Crandall et al. 2000) and that integrative taxonomic approaches, combining molecular, ecological, and behavioral data, offer a more holistic perspective on evolutionary significance. The study would benefit from broader geographic coverage, particularly from Mongolian populations, which are genetically close but exhibit some differentiation (FST = 0.67). Longitudinal and cross-border sampling would be especially valuable for understanding historical and contemporary gene flow.

I also recommend elaborating about how habitat fragmentation and anthropogenic barriers may create patterns mimicking long-term isolation.

Author Response

(The authors gave the same response as above.)

Round 2

Reviewer 2 Report

Comments and Suggestions for Authors

Authors addressed most of the raised criticism related to the definition of ESU.

However, a couple of major points still require attention.

In the genetic analyses, authors evaluate the ratio of transitions (Ti) to transversions (Tv) but then they make no use of it. What did they use it for?

I could not find any specification about the methods used by the authors when reconstructing the phylogeny. Also, there is some confusion if the representation of the trees in figs.2 and 4. In fact, the tree in figs 2 are cladograms as they report no information about branch length. The only tree that reports branch length is in fig. 4. I think that all trees are constructed based on “genetic distance”, but no information is provided about what genetic distance is being used.

If the goal of the authors is investigating topology, with no interest in reconstructing a phylogram or a chronogram, they should at least seek statistical support to the existence of separate clades. In other words, they should care to add a node validation procedure, such as bootstrapping.

Anyway, the analysis can be summarized in a single concatenated tree, in only one figure.

Separate analyses could be reported in the Supplementary

This part must be scrutinized, before proceeding further.

Author Response

We sincerely thank you for your thoughtful and constructive comments on our manuscript titled “Genetic Diversity and Phylogenetic Relationships of Castor fiber birulai in Xinjiang, China, Revealed by Mitochondrial Cytb and D-loop Sequence Analyses.” animals-3718123;We are grateful for your positive evaluation of our work and your helpful suggestions to improve the manuscript. Below we provide a point-by-point response to each of your comments. All revisions have been incorporated into the revised manuscript, and changes are marked accordingly.

Reviewer’s comment 1:

In the genetic analyses, authors evaluate the ratio of transitions (Ti) to transversions (Tv) but then they make no use of it. What did they use it for?

Response 1:

We sincerely thank the reviewer for pointing out the lack of explanation regarding the use of the transition/transversion (Ti/Tv) ratio and the phylogenetic reconstruction methods.

In response, we have revised the Materials and Methods section to clarify these aspects. Specifically, we have now stated that the Ti/Tv ratio was calculated using MEGA 11.0 to evaluate mutational bias and to inform the selection of appropriate nucleotide substitution models.These revisions can be found on page 6, lines 170–172 of the revised manuscript.

Reviewer’s Comment 2:

The study is limited by the use of only mitochondrial DNA, which does not capture male-mediated gene flow. Including nuclear DNA markers (such as microsatellites or SNPs) would enhance understanding of population structure and history.

Response 2:

We fully agree with your valuable suggestion. We have also provided detailed information about the phylogenetic reconstruction procedures. Separate Bayesian inference (BI) trees were generated for both Cytb and D-loop datasets, as well as a concatenated tree based on the combined dataset. These analyses were performed in MrBayes v3.2.7 under the GTR+I+G model, with parameters accounting for substitution rate variation, base frequency, the Ti/Tv ratio, invariant sites, and rate heterogeneity among sites. The analyses were run for 1,000,000 generations with sampling every 100 generations, and the first 25% of samples were discarded as burn-in to obtain a stable posterior distribution.

These revisions can be found on page 6, lines 172–179 of the revised manuscript.

Reviewer’s Comment 3:

If the goal of the authors is investigating topology, with no interest in reconstructing a phylogram or a chronogram, they should at least seek statistical support to the existence of separate clades. In other words, they should care to add a node validation procedure, such as bootstrapping.

Response 3:

Thank you for this valuable suggestion. Although our analysis focuses on phylogenetic topology, we agree that statistical support for clades is essential. As our phylogenetic reconstruction was performed using Bayesian inference in MrBayes, we have included posterior probability values at all internal nodes in the revised tree (Figure 2) to indicate clade support. Posterior probabilities are a widely accepted alternative to bootstrapping for assessing node confidence in Bayesian analyses.

While the node support values are labeled as “bootstrap” in the figure due to iTOL software display conventions, we clarify that these values represent Bayesian posterior probabilities, not resampling-based bootstrap values. This clarification has been added to the figure legend and Methods section.

Reviewer’s comment 4:

Anyway, the analysis can be summarized in a single concatenated tree, in only one figure.

Response 4:

We sincerely thank the reviewer for the thoughtful and constructive suggestion. As advised, the original Figure 4 (the phylogenetic tree based on D-loop sequences) has been moved to the Supplementary Materials and renamed as Figure S2 Phylogenetic Tree Based on D-loop Genetic Distances among Different Castor fiber Subspecies.” We believe this revision improves the clarity and organization of the manuscript, and we truly appreciate your careful review and valuable input.Separate analyses could be reported in the Supplementary.
    We sincerely thank the reviewer for their thoughtful and thorough review of our manuscript. Your constructive comments have not only helped us identify areas needing improvement but have also guided us to strengthen the overall quality and clarity of our research. We deeply appreciate the time, effort, and expertise you have devoted to this review, and we are truly grateful for your valuable contribution.

Sincerely,
Linyin Zhu
Corresponding Author on behalf of all co-authors

Xinjiang University  

zhujulien00@gmail.com

Date: 10 July 2025
